# Enhanced Development of Sweat Latent Fingerprints Based on Ag-Loaded CMCS/PVA Composite Hydrogel Film by Electron Beam Radiation

**DOI:** 10.3390/gels8070446

**Published:** 2022-07-18

**Authors:** Jinyu Yang, Yayang Wang, Yuan Zhao, Dongliang Liu, Lu Rao, Zhijun Wang, Lili Fu, Yifan Wang, Xiaojie Yang, Yuesheng Li, Yi Liu

**Affiliations:** 1Key Laboratory of Coal Conversion and New Carbon Materials of Hubei Province, School of Chemistry and Chemical Engineering, Wuhan University of Science and Technology, Wuhan 430081, China; yjyxjj@wust.edu.cn (J.Y.); 1750934313@163.com (Y.W.); zhyf308@hbust.edu.cn (Y.Z.); 2Hubei Key Laboratory of Radiation Chemistry and Functional Materials, Non-Power Nuclear Technology Collaborative Innovation Center, Hubei University of Science and Technology, Xianning 437100, China; ldl142325@163.com (D.L.); rl13908440364@163.com (L.R.); qwertasdkl@163.com (Z.W.); fll151852538462022@163.com (L.F.); wang965265@126.com (Y.W.); mailyangxiaojie@126.com (X.Y.); 3College of Chemistry and Chemical Engineering, Tiangong University, Tianjin 300387, China

**Keywords:** Ag-loaded CMCS/PVA hydrogel film, sweat latent fingerprints, development, electron beam radiation, cytotoxicity

## Abstract

Over time, difficulties have been encountered in detecting potential fingerprints. In this study, an Ag/CMCS/PVA(ACP) hydrogel film was developed for fingerprint development by electron beam radiation method. The chemical bond, thermostability, chemical components, microstructure, and micromorphology of the CMCS/PVA composite hydrogel film were characterized by FT-IR, TG, XRD, and SEM, respectively. Swelling behaviors and mechanical performance of the CMCS/PVA composite hydrogel were also investigated at different irradiation doses, pH, media, and NaCl contents to obtain the optimum preparation conditions. Through experimental exploration, we found that the fingerprints appeared more obvious when the irradiated prepared ACP hydrogel film was sprayed with 0.6 mg/mL of Ag^+^ and the excitation wavelength was about 254 nm with UV lamp irradiation for 20 min. The cytotoxicity the CMCS/PVA composite hydrogel on mouse skin fibroblasts L929 cells was also studied, confirming its biological security. Sweat latent fingerprint manifestation has important scientific significance with respect to the development of new processes and functional materials in the field of fingerprint manifestation, enriching and complementing the application of composite hydrogels.

## 1. Introduction

Human fingerprints are used as the primary means of identification in forensic investigation because they have two advantages [1]: (1) no two people have the same fingerprint, and even identical twins have different fingerprints; and (2) the ridge-and-valley pattern of fingerprints does not change with age. Fingerprints can be classified as formed fingerprints, obvious fingerprints, and latent fingerprints. Finding visible or molded fingerprints at a crime scene is a relatively simple task, as they are usually visible to the naked eye [2]. However, latent prints refer to fingerprints that are not observed. Potential sweat prints are often found at crime scenes on objects that are frequently handled or accidentally touched during the course of a crime. Potential sweat fingerprinting requires the development of methods that enable visualization and recording of fingerprints [3,4]. Therefore, it is essential to choose an appropriate approach for visualization of latent prints. This selection depends on the surface to be inspected, ease of use, effectiveness, and efficiency, as well as health considerations [5]. In the past, physical development (small-particle reagent, fumigation, and vacuum metal deposition), chemical development (silver nitrate, ninhydrin, and its analogs), light development (laser, fluorescence, UV/vis, and infrared absorption), and combined techniques have been explored for visualization or enhancement of latent fingerprints in specific situations [6]. To date, few studies have been conducted on the preparation of CMCS/PVA composite hydrogel films for enhanced development of sweat latent fingerprints.

Due to their three-dimensional, hydrophilic, polymeric network, hydrogels are able to absorb large amounts of water or biofluids [7]. Smart hydrogels have attracted extensive attention, owing to their ability to respond to physical and chemical stimuli in the external environment, such as temperature [8,9], pH [10], magnetic field [11], ionic strength [12], electrical field [13], photoirradiation [14,15], etc. Hydrogels have been used extensively in the field of drug controlled release [16], sensors [17], tissue engineering [18], drug control [19], etc. Chitosan is widely distributed in nature as a renewable polymer. It can be used as a natural dressing, and its degradability and biocompatibility have received a considerable amount of attention [20]. Compared with other chitosan derivatives, carboxymethyl chitosan (CMCS) has been broadly studied because of its easy synthesis [21,22]. Poly(vinyl alcohol) (PVA) is used widely for biomedical [23] and biochemical applications [24] due to its excellent properties, such as high water content, low toxicity, biocompatibility, good mechanical properties, and hydrophilicity. Varaprasad et al. [25] introduced nano-Ag into a polyacrylamide/polyvinyl alcohol composite gel and found that the nanocomposite hydrogel can be used to effectively heal wounds. Xu et al. [26] synthesized pH-sensitive Ag/AgO/carboxymethyl chitosan bacteriostatic hydrogels and drug carriers by in situ precipitation.

Electron beam radiation is a simple method of hydrogel preparation that is particularly attractive for future scale-up production [27]. The radiation induction process has many unique advantages over commonly used chemical methods, such as high efficiency, no requirement for an initiator, no need for intelligent control of experimental conditions, and no restrictions on the shape of the backbone polymer [28]. In recent years, the radiation method has been widely used for the preparation of dressings [29,30], drug delivery [31,32,33], photodegradable materials [34], antimicrobial materials [35], and other hydrogels. Whether hydrogel and radiation methods can be combined effectively and used in the field of fingerprint development is worthy of attention. Therefore, the goal of the present study is to successfully synthesize ACP composite hydrogel films by in situ electron beam radiation method and apply them to effectively enhance the development of sweat latent fingerprints.

Although existing fingerprint examination techniques have reached a stage of ease of operation, accuracy, and reliability, examination of fingerprints using hydrogel films has not yet been reported. In this work, ACP hydrogel films were successfully prepared by electron beam radiation method for the development of sweat latent fingerprints. We systematically investigated swelling behaviors at different irradiation doses, pH, media, NaCl content, as well as developing effects with different excitation wavelengths and irradiated times with a UV lamp, preparing methods, coating modes, concentrations of Ag^+^, types of developing reagents, etc. The chemical bond, thermostability, chemical components, microstructure, and micromorphology of the CMCS/PVA composite hydrogel films were characterized by FT-IR, TG, XRD, and SEM, respectively. The cytotoxicity of the CMCS/PVA composite hydrogel on mouse skin fibroblasts L929 cells was also studied to confirm its biological security. APC hydrogel film is scientifically important for sweat latent fingerprint development, enriching and complementing the application of composite hydrogels in the field of biofunctional materials, such as fingerprint development materials.

## 2. Results and Discussion

### 2.1. Swelling Behaviors of CMCS/PVA Hydrogel Film at Different Conditions

The swelling properties of CMCS/PVA hydrogel are important for sweat latent fingerprint development film to obtain clearer images. Varying swelling behaviors were observed under different conditions (Figure 1). The swelling degree first increased and then decreased with the irradiation dose (Figure 1A). The swelling degree of the hydrogel was the highest when the irradiation dose was 30 kGy. The gel fraction curve first increased with the CMCS/PVA mass ratio and then decreased (Figure 1B) when the mass ratio was 3:1, the highest gel content, indicating an improved crosslinked degree of the hydrogel. However, the swelling degree was the highest with a mass ratio of 4:1 (Figure 1C). According to the relationship between pH and swelling degree (Figure 1D), the swelling degree of hydrogel increases with increasing pH value when the pH is less than 4; however, when pH is greater than 4, the of the swelling degree of the hydrogel is slowed down. CMCS/PVA hydrogel exhibits different swelling behavior in other media solutions. We observed the maximum degree of swelling of the hydrogel in a pH = 9 environment (Figure 1E). Figure 1F shows that the swelling degree of CMCS/PVA hydrogel can decreased increased NaCl content. These swelling properties of CMCS/PVA hydrogel under different conditions provide a solid reference for preparing high-quality sweat latent fingerprints development film.

### 2.2. Mechanical Properties of CMCS/PVA Hydrogel Film

As freezing–thawing progressed, the tensile strength and Young’s modulus of the CMCS/PVA hydrogel increased, as shown in Figure 2. From the third freeze–thaw cycle to the seventh cycle, the Young’s modulus of the hydrogel increased by 1.7 MPa (Figure 2A), indicating that freezing–thawing can enhance the Young’s modulus of a hydrogel. According to the modulus–strain relationship shown in Figure 2B, the tensile modulus of the hydrogel varies significantly depending on the region. In the range of 0~100%, the tensile modulus of the hydrogel is almost constant, exhibiting elasticity. Stress–strain performance is linear, but when the content was between 100% and 250%, the tensile modulus of the hydrogel increased with increasing strain, exhibiting typical viscoelasticity and stress–strain. The tensile modulus of the hydrogel gradually stabilized with increasing pressure beyond 250%. These mechanical properties of hydrogel provide a guarantee for later molding and sweat latent fingerprints development applications.

### 2.3. Microstructural Studies

Figure 3A shows the IR spectra of PVA, CMCS, and the CMCS/PVA hydrogel sample. The major peaks occur at 3365 cm^−1^ (O-H stretching), 2906 cm^−1^, 2852 cm^−1^ (C-H stretching), 1150 cm^−1^ (C-O stretching of PVA), 1417 cm^−1^ (O-H deformation), and 1244 cm^−1^ (C-O stretching of PVA). The strong peaks were assigned to -OH stretching of adsorbed bound water (3433.26 cm^−1^) and the -NH_2_ absorption peak (1086.76 cm^−1^) from the spectrum of the CMCS group. Similarly, the strong characteristic peaks of -OH at 3338 and 1422 cm^−1^ were assigned to -OH stretching vibration, and CH-OH bending vibration absorption peaks were also observed in the spectrum of the PVA group. The infrared characteristic peaks of PVA/CMCS hydrogel film were roughly the same as those of PVA and CMCS, without chemical changes.

The XRD spectra are shown in Figure 3B. CMCS and PVA also show narrow bands at 2*θ* = 20°, demonstrating their crystalline structure. CMCS/PVA hydrogel has a more enhanced peak width and peak height at different 2*θ*, with similar peak shapes to those of a single PVA. These results indicate that the original crystal structures of CMCS and PVA were altered after the electron beam radiation crosslinking reaction.

### 2.4. TG Analysis

The thermal stability of single CMCS, PVA, and CMCS/PVA hydrogels was determined by thermal gravimetric analysis. Figure 3C shows thermal gravimetric images of the above three samples, with slight weight loss under 210 °C was owning to the evaporation of free water and bound water in the samples. However, the weight loss rate was accelerated at temperatures from 210 °C to 500 °C, which indicates the decomposition of CMCS and PVA at temperatures higher than 210 °C. CMCS and PVA were decomposed with the melting process and completely decomposed and carbonized at temperatures higher than 500 °C. Thermal gravimetric analysis curves of the one-component PVA and the CMCS/PVA crosslinked composite hydrogel exhibited no significant differences. Overall, PVA exhibited the most considerable weight loss, followed by the CMCS/PVA composite, because some crosslinked microstructures in the CMCS/PVA composite hydrogel change its thermal stability.

### 2.5. Morphology

Figure 3D, E show the micromorphology of the CMCS/PVA composite hydrogel, with some holes and gaps in the surface structure of the hydrogel. In the higher-resolution images (Figure 3F,G), the regular skeleton and the uniform distribution of the transverse hole of the hydrogel can be observed. The hydrogel has a powerful swelling capacity because of high internal porosity. Furthermore, the presence of internal pores in the hydrogel, which led to the its high surface area, is significant in the field of intelligent hydrogels.

### 2.6. Effect of Varying UV Light Wavelengths

It is difficult to observe clear sweat latent fingerprints on ACP hydrogel films with the naked eye under ordinary natural light. Therefore, varying excitation wavelengths were used with a UV lamp (254 nm, 356 nm), and the excitation wavelength dependence was examined to obtain enhanced sweat latent fingerprint development hydrogel films (Figure 4A,B). Figure 4A shows that an improved developing effect on sweat latent fingerprints was achieved at the excitation wavelength of 254 nm under a UV lamp. Therefore, subsequent experiments were conducted with an excitation wavelength of 254 nm to further investigate other factors.

### 2.7. Influence of Irradiation Time

Different irradiation times for the decomposition of AgCl are presented in Figure 4C–G, showing that sweat latent fingerprints can be developed with higher definition with increasing the irradiation times. However, a considerable amount of AgCl spots are present after more than 20 min, leading to a reduction in the clarity of potential sweat fingerprints. Therefore, the optimal irradiation time to develop the images is 20 min.

### 2.8. Relationship between Preparation Method and Fingerprint Development

Three preparation methods were tested to obtain high-quality ACP hydrogel sweat latent fingerprint development films (Figure 4H–J). The sweat latent fingerprint image produced by freezing without irradiation is not clear (Figure 4H), most likely due to the coalescence of AgCl spots and the microstructure of the hydrogel film affecting fingerprint development. Similarly, Figure 4I shows the relative obscuring effect of refrigeration with irradiation. Finally, the method of short-term freezing and irradiation was adopted to successfully prepare the clearest image with ACP hydrogel sweat latent fingerprint development (Figure 4J), with a suitable microstructure obtained by this method.

### 2.9. Impact of Different Coating Modes

Ag^+^ must effectively and develop clearly into a CMCS/PVA hydrogel film. Three loading methods are shown in Figure 4: such as spray (Figure 4K), smear (Figure 4L), and soak (Figure 4M). The spray method has a better developing effect than that of the smear and soak methods. Several reasons could explain the inferior images in Figure 4L,M: First, insufficient or excessive smearing leads to uneven distribution of Ag^+^. Second, the spot method of may cause damage to the original fingerprint. Finally, it is difficult to control the Ag^+^ concentration to ensure a good sweat latent fingerprint development effect using the smear and soak methods. Therefore, the spray-on mode of Ag^+^ coating preferable.

### 2.10. Behaviors of Fingerprint Development with Different Concentrations of Ag^+^ Ions

As described above, the appropriate concentration of Ag^+^ has a significant effect on sweat latent fingerprint development with CMCS/PVA hydrogel films. Varying concentrations of Ag^+^ were tested: 0.12 mg/mL, 0.6 mg/mL, 1.2 mg/mL, 1.8 mg/mL, and 2.4 mg/mL. As shown in Figure 4N–R sweat latent fingerprints are not visible at the low concentration of 0.12 mg/mL (Figure 4N). Relatively clear fingerprints can be observed with a concentration of 0.6 mg/mL (Figure 4O), whereas the fingerprints are broken or stacked with increasing concentrations of Ag^+^. Therefore, according to experimental results, 0.6 mg/mL was selected as the optimum Ag^+^ concentration.

### 2.11. Comparison of Fingerprint Development with Different Developing Reagents

To demonstrate that CMCS/PVA hydrogel can be widely used with a variety of developing reagents, we compared ninhydrin to Ag^+^ as a developing agent. Ninhydrin, a common chemical reagent, is widely used in forensic science. An amino acid of fingerprint residue that produces a purple print following stoichiometric reaction was reported by Ruhemann [36,37], and the reaction pathway of latent fingerprint could also be visualized. CMCS/PVA hydrogel film has an excellent developing effect when used with both of the investigated agents, as shown in Figure 4S,T. Furthermore, CMCS/PVA hydrogel film can be widely used with a wide variety of developing reagents.

### 2.12. Evaluation of the Toxicity of Hydrogel Film by the Alamar Blue Method

Alamar blue serves as a non-fluorescent blue–purple redox indicator under oxidation conditions [38]. It can be converted to pink or red fluorescent reduction products in a reduced state to indicate the metabolic activity of cells [39]. The Alamar blue is released to the cell and dissolved in the culture medium after the reduction of the culture medium is revealed. Because the fluorescence intensity is proportional to the number of active cells, the reduction rate of Alamar blue can be calculated from the fluorescence value, effectively reflecting the cellular activity and the interfering factors with respect to cytotoxicity.

Data with respect to the effect of varying concentrations of hydrogels on the activity of L929 cells are shown in Table 1. With increased concentration of hydrogel extracting solution (from 0.1 g/mL to 0.4 g/mL), the three different types of hydrogels exhibited no apparent cytotoxicity on L929 cells, with no significant effect on the reduction of Alamar blue. The impact of other hydrogel extracting solutions on the morphologies of L929 cells is shown in Figure 5, which was also confirmed by the experimental results, indicating that the CMCS/PVA composite hydrogel film had no apparent toxicity on L929 cells.

### 2.13. Mechanism of Fingerprint Development

Sweat is secreted by the pores of the fingerprint ridge, and the fingerprint curve is formed by finger pressure on the CMCS/PVA composite hydrogel. The mechanism of silver nitrate fingerprints is as follows: first, the sodium chloride in sweat combines with silver nitrate to produce silver chloride; then, the fingerprint film is irradiated under excitation light, which causes the silver chloride to decompose into black-brown silver particles, thus revealing the fingerprint (Figure 6) [40].

## 3. Conclusions

A new type of enhanced development of sweat latent fingerprints based on ACP hydrogel film was successfully prepared by in situ electron beam radiation. CMCS/PVA composite hydrogel film was characterized by FT-IR, TG, XRD, and SEM to analyze its microstructure and micromorphology. To obtain the best physical and chemical properties for later enhanced sweat latent fingerprint development, swelling behaviors and mechanical performance were also discussed. We obtained the optimum preparation conditions of CMCS/PVA composite hydrogel under different irradiation doses, pH, media, and NaCl contents. Through experimental exploration, we found that the fingerprints appeared more obvious when the irradiated prepared ACP hydrogel film was sprayed with 0.6 mg/mL of Ag^+^ and the excitation wavelength was about 254 nm with UV lamp irradiation for 20 min. The cytotoxicity of CMCS/PVA composite hydrogel on mouse skin fibroblast L929 cells was also studied, confirming its biological security. Sweat latent fingerprint manifestation has paramount scientific significance with respect to the development of new processes and functional materials in the field of fingerprint manifestation, enriching and complementing the application of composite hydrogels.

## 4. Experimental

### 4.1. Materials

Carboxymethyl chitosan (CMCS), polyvinyl alcohol (PVA, 1750 ± 50), ninhydrin, and AgNO_3_ were analytically pure (Sinopharm Chemical Reagent Co., Ltd., Shanghai, China). Alamar blue, 1640 fetal bovine serum, peptone, yeast extract, and beef extract, were purchased from Sigma Company (Shanghai, China). L929 cells were provided by Wuhan University.

### 4.2. Preparation of CMCS/PVA Hydrogel Film by Electron Beam Radiation

CMCS/PVA hydrogel film was prepared as follows: A given mass of PVA particles was added to deionized water and heated to 95 °C under magnetic stirring for 2 h to prepare a 10% PVA aqueous solution. Then, 0.5 g CMCS was added to the PVA aqueous solution under magnetic stirring for 0.5 h to fully dissolve the CMCS, with a mass ratio of CMCS: PVA of 1:4. After the solution was thoroughly mixed and added to a 60 mm culture dish, the whole system (culture dish and CMCS/PVA solution) was put in a freezer for 2 h to obtain a 3 mm thick film. Finally, the samples were irradiated at doses of 15–35 kGy and at a dose rate of 5 kGy/pass by a 1 MeV electron accelerator (voltage of 750 kV, beam current of 10 mA). The samples were soaked in deionized water for 24 h and then backed-up under refrigeration.

### 4.3. Preparation of CMCS/PVA Hydrogel Sweat Latent Fingerprint Film

The test procedure was designed to confirm whether new fingerprints from healthy donors were deposited on CMCS/PVA composite hydrogel film. To ensure consistency, we collected natural fingerprints from donors, which are known to produce good natural traces. Before each fingerprint application, the donor’s finger was cleaned with soap and water and exposed to dry air for 30 min. Then, the donors gently wiped their forehead with their fingers. Finally, donor fingers were pressed on the surface of the CMCS/PVA composite hydrogel film to obtain sweat latent fingerprints at room temperature in an air atmosphere.

### 4.4. Development of ACP Hydrogel Sweat Latent Fingerprint Film

Because of sweat, latent fingerprints on the ACP hydrogel film were invisible to the naked eye under ordinary natural light. To identify sweat latent fingerprints, different excitation wavelengths were tested with UV lamps (254 nm, 356 nm), and the excitation wavelength dependence was examined. Images of the fingerprints were taken with a digital camera (Sony D80). During the experiments, the temperature was kept between 25 °C and 30 °C, and the relative humidity was maintained at 65–70%.

To prepare for quick and improved variation of ACP hydrogel sweat latent fingerprints films, the experiment was executed for one h under 254 nm UV light, and the results of the development of sweat latent fingerprints were observed with different irradiated times, preparation methods, and coating modes. An appropriate concentration of Ag^+^ ions is essential to achieve high-quality development of sweat latent fingerprints. The effect of the concentration of Ag^+^ ions was investigated and identified as 0.12 mg/mL, 0.6 mg/mL, 1.2 mg/mL, 1.8 mg/mL, and 2.4 mg/mL. We also compared experimental results obtained with ninhydrin and Ag^+^ ions to confirm that CMCS/PVA hydrogel sweat latent fingerprint film can be used widely with multiple developing agents.

### 4.5. Gel Fraction

The gel fraction (*GF*) of the hydrogel extracted in distilled water at 80 °C for 36 h, then dried at 60 °C for 48 h until a constant weight was reached. The gel content of the hydrogel was determined as follows:(1)$$GF\left(\%\right)=\frac{W_{e}}{W_{d}}\times 100\%\;$$
where *W_d_* and *W_e_* denote the weight of the dried hydrogel and gel fraction, respectively.

### 4.6. Swelling Behaviors

The swelling rate (*SR*) was measured by soaking the dried hydrogel (*W_d_*) in an excess of distilled water at room temperature for 36 h until equilibrium. Then, the mass of the wet hydrogel (*W_t_*) was measured after removing the surface water. The swelling ratio of the hydrogels was calculated with the following equation:(2)$$SR=\frac{W_{t}-W_{d}}{W_{d}}$$

### 4.7. Characterization

Fourier transform infrared spectra (FTIR) were recorded on a NICOLET 5700 spectrometer (Thermo Fisher Nicolet, Mandison, WI, USA) in the range of 4000–400 cm^−1^. Thermal gravimetric analysis was carried out with a NETZSCH thermogravimetric analyzer (TG209F3, Selb, Germany). The dried samples were put in a crucible and heated from 30 °C to 700 °C at a heating rate of 10 °C min^−1^ in nitrogen. The crystal structures of the products were characterized by X-ray powder diffraction (DMAX-D8X, Rigaku, Japan). The patterns with Cu-Kα radiation (λ = 0.15406 nm) at 20 kV and 10 mA were recorded in the 2θ range of 10° to 80° (2θ) with a scan speed of 4° min^−1^. The morphology and microstructure were characterized by scanning electron microscopy (VEGA3-SBH, Tescan, Czech Republic), and all the samples were coated with Au on a universal testing machine (AG-IC, 20 KN/50 KN Shimadzu, Kyoto, Japan) Triusage Ultraviolet Analysis Instrument (ZF-1, Shanghai Chi Tang Electronics Co., Ltd., Shanghai, China). Fluorescent spectrophotometer (960 MC/CRT, Shanghai). Biological microscope (XSP-2CA, Shanghai, China). 1 MeV electron accelerator (voltage of 1000 kV, beam current of 50 mA, Wasik Associates, Dracut, MA, USA).

### 4.8. Cell Toxicity Test

Cell toxicity testing was performed as described by the Li group [35] as follows: Dried and mashed hydrogels (2 g, 4 g, and 8 g) were added to a 20 mL 1640 culture medium containing 10% fetal bovine serum and incubated at 37 °C under 5% CO_2_, then extracted by 0.22 µm membrane filtration sterilization. The cell concentration was adjusted to 1 × 10^4^ mL^−1^ and added to 96-well plates and incubated in a box for 24 h at 37 °C under 5% CO_2_ to obtain the logarithmic growth period of L929 cells. A leaching solution of hydrogel was added to 96-well plates. Each group was divided into three complex wells and incubated in a box at 37 °C under 5% CO_2_ for 24 h. A volume of 10 µL Alamar blue solution was added to each well when the medium color changed from cyan blue to pink. The fluorescence intensity of the samples was detected by spectrophotometer, and the Alamar blue reduction rate was calculated.

## Figures and Tables

**Figure 1 gels-08-00446-f001:**
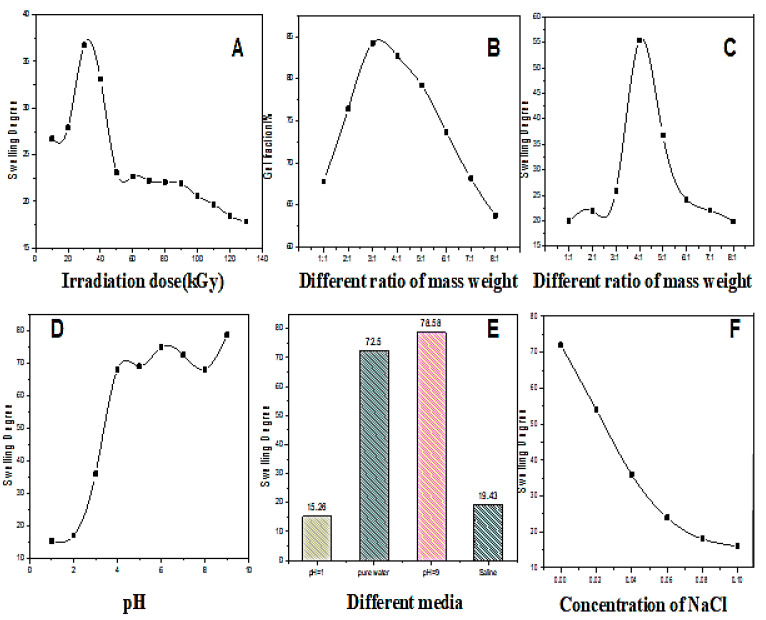
Swelling behavior of CMCS/PVA hydrogel under different conditions. (**A**) Swelling degree curves with varying irradiation doses; (**B**) gel fraction curves with varying ratios of mass weight; (**C**) swelling degree curves with varying ratios of mass weight; (**D**) swelling degree curves with varying pH values; (**E**) swelling degree curves in different media; (**F**) swelling degree curves with varying concentrations of NaCl solution. T = 298 K; irradiation dose = 30 kGy.

**Figure 2 gels-08-00446-f002:**
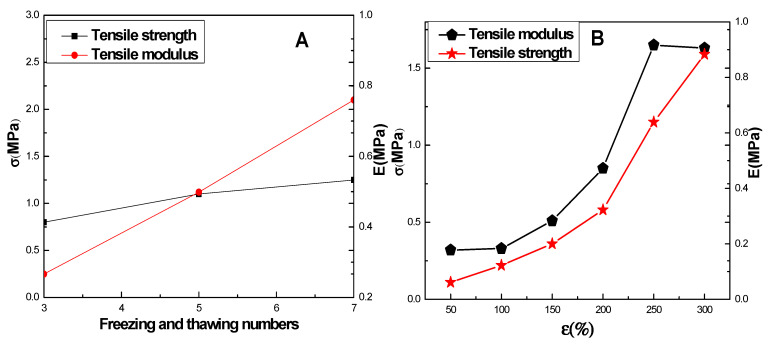
Tensile strength and Young’s modulus of CMCS/PVA hydrogels. (**A**) Tensile strength and modulus with varying numbers of freezing–thawing cycles; (**B**) variation law of stress and instantaneous tensile modulus with strain.

**Figure 3 gels-08-00446-f003:**
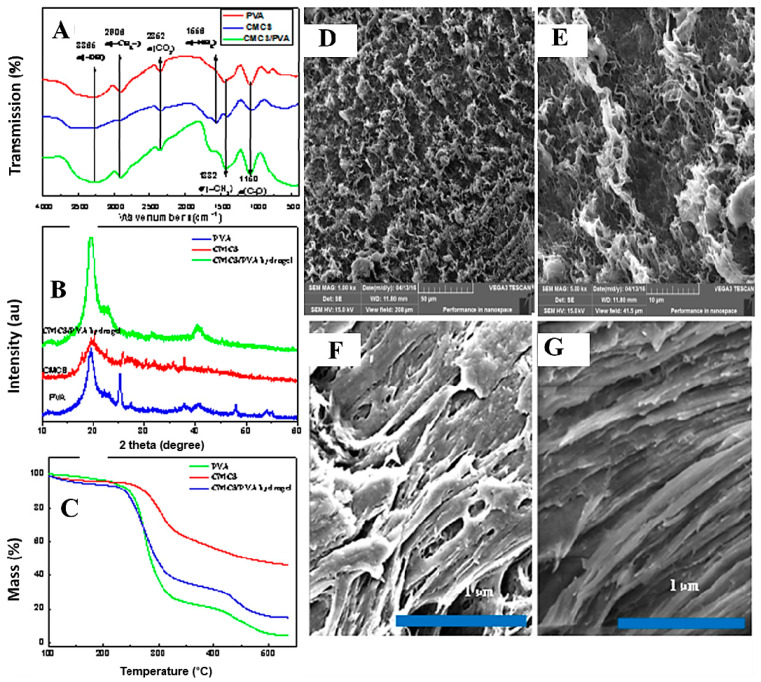
Characterization of the microstructure and morphology of the CMCS/PVA composite hydrogel films. (**A**) FT-IR; (**B**) XRD; (**C**) TG; (**D**–**G**) SEM.

**Figure 4 gels-08-00446-f004:**
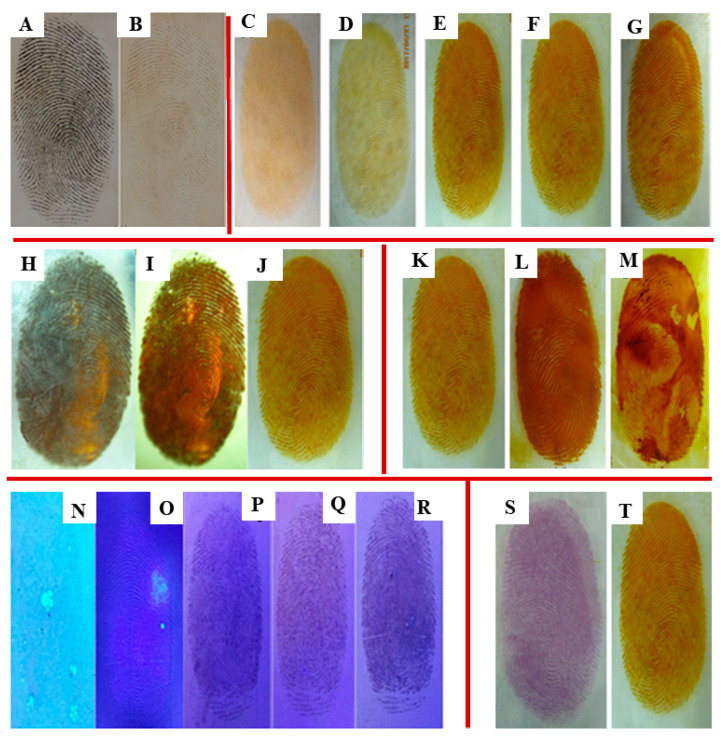
Fingerprint developing photos of CMCS/PVA composite hydrogel films under different external conditions: with excitation wavelengths of 254 nm (**A**) and 356 nm (**B**); at different irradiated times under 254 nm UV light: (**C**) 5 min, (**D**) 10 min, (**E**) 20 min, (**F**) 30 min, (**G**) 40 min; with different preparation methods: (**H**) frozen without irradiation, (**I**) refrigeration and irradiation, (**J**) short-term freezing and irradiation; with different coating modes: (**K**) spray, (**L**) smear, (**M**) soak; with different concentrations of Ag ions: (**N**) 0.12 mg/mL, (**O**) 0.6 mg/mL, (**P**) 1.2 mg/mL, (**Q**) 1.8 mg/mL, and (**R**) 2.4 mg/mL; with different developing reagents: (**S**) ninhydrin, (**T**) Ag ions.

**Figure 5 gels-08-00446-f005:**
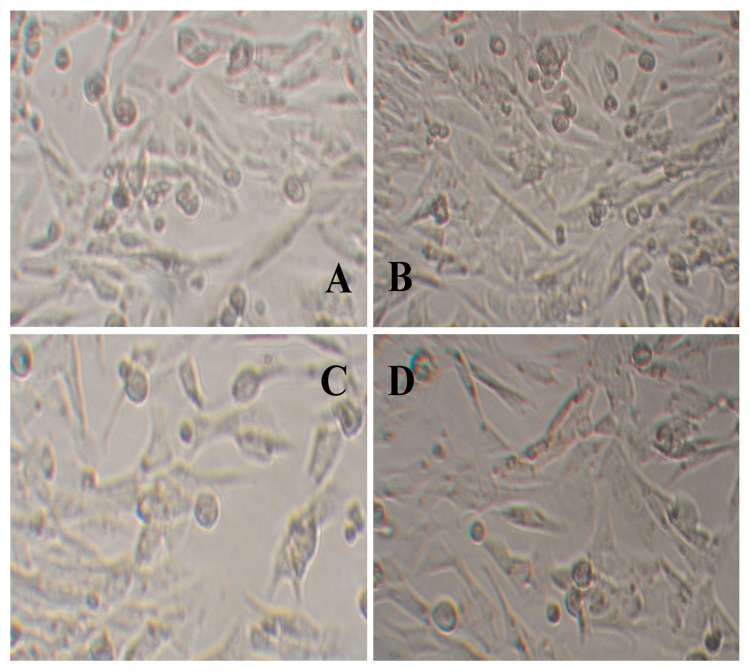
Images of cell growth of L929 with the addition of different hydrogel leaching solutions. (**A**,**B**) No CMCS/PVA hydrogel leaching solution added; (**A**) before, (**B**) after. (**C**,**D**) CMCS/PVA hydrogel leaching solution added; (**C**) before, (**D**) after.

**Figure 6 gels-08-00446-f006:**
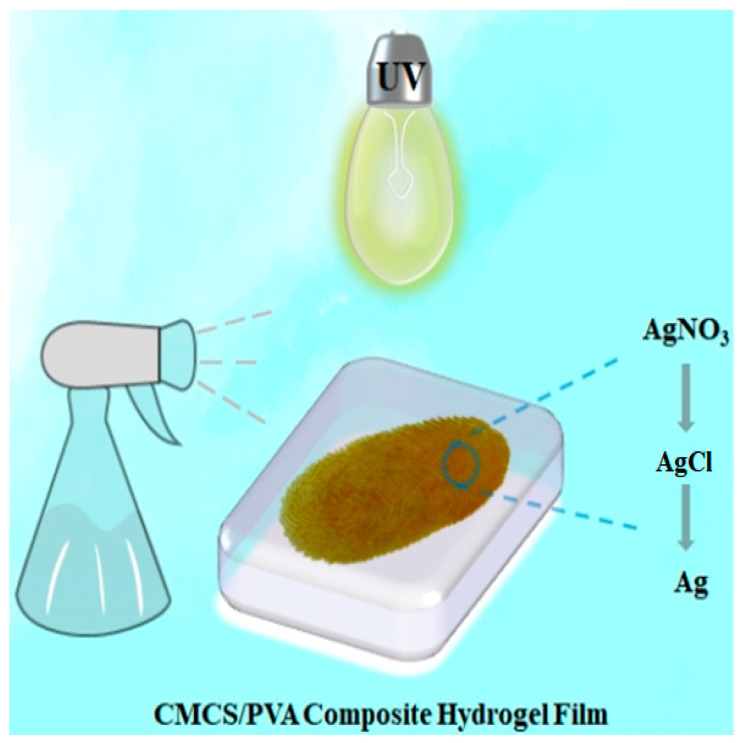
Schematic diagram of hydrogel fingerprint film development mechanism.

**Table 1 gels-08-00446-t001:** Effect of hydrogel leaching solution concentration on L929 cell activity.

Sample	Alamar Blue Reduction Ratio (%)		
	0.1 g/mL	0.2 g/mL	0.4 g/mL
Control	76.12 ± 5.22	75.91 ± 5.38	75.52 ± 5.36
CMCS	76.09 ± 2.35	73.96 ± 3.52	72.86 ± 1.75
PVA	75.18 ± 2.74	73.93 ± 4.01	72.06 ± 1.55
CMCS/PVA hydrogel	74.38 ± 3.54	72.99 ± 4.77	70.28 ± 3.51

## Data Availability

Not applicable.

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
