# Peer review of "Enhanced Development of Sweat Latent Fingerprints Based on Ag-Loaded CMCS/PVA Composite Hydrogel Film by Electron Beam Radiation"

_gels, 2022, doi:10.3390/gels8070446_

Round 1
Reviewer 1 Report
The manuscript entitled 'Enhanced Development of Sweat Latent Fingerprints Based on Ag Loaded CMCS/PVA Composite Hydrogel Film by Electron Beam Radiation' by Yang et al. describes an electron beam radiation method for the preparation of Ag-loaded-CMCS/PVA (ACP) hydrogel film for sweat latent fingerprints development. The same was characterized well by various techniques such as FT-IR, TG, XRD, and SEM. The manuscript is well written. Authors have developed new processes and functional materials in the field of fingerprint manifestation. Therefore, the present reviewer thinks this manuscript can be accepted in this journal.
Author Response
Dear reviewer,
Thank you very much for your support of this article.
Reviewer 2 Report
The article 'Enhanced Development of Sweat Latent Fingerprints Based on Ag Loaded CMCS/PVA Composite Hydrogel Film by Electron Beam Radiation' describes the use of hydrogels for sweat latent fingerprint manifestation. This is an interdisciplinary work which is interesting for the general public. The scientific level of the article meet the requirements of the Gels journal, and the manuscript can be accepted after minor revision.
The Abstract should be revritten according to Gels' Instructions for Authors: 'The abstract should be a single paragraph and should follow the style of structured abstracts, but without headings: 1) Background: Place the question addressed in a broad context and highlight the purpose of the study; 2) Methods: Describe briefly the main methods or treatments applied... 3) Results: Summarize the article's main findings; and 4) Conclusion: Indicate the main conclusions or interpretations.'
Minor remarks:
p. 3 – 3: 1, 4: 1 – remove spaces before ':'
p. 3 and below – do not use abbreviations like 'Fig 1'
p. 3 – Figure 1 seems stretched vertically
p. 3 and below – remove blue frames around figures
p. 4 and below – remove spaces before '%'
p. 4 and below – use '-' and '–' correctly
p. 11 and below – it is necessary to specify reagent or equipment suppliers (Company, City, State Abbreviation, Country).
Author Response
Please consider as attachment.

Reviewer 3 Report
In this work, the authors investigate the Enhanced Development of Sweat Latent Fingerprints Based on Ag Loaded CMCS/PVA Composite Hydrogel Film by Electron Beam Radiation. Some corrections are required for this manuscript to be accepted.
It is recommended that the authors focus more on the biocompatibility of hydrogels in the introduction part of the manuscript. This part should be given in more detail, and the literature should be used. You can cite the following articles for this section of your manuscript.
1. A review on biocompatibility nature of hydrogels with 3D printing techniques, tissue engineering application and its future prospective. https://doi.org/10.1007/s42242-018-0029-7.
2. Biocompatibility of PEG-Based Hydrogels in Primate Brain, https://doi.org/10.3727/096368908784423292.
3. Origin of the synthetic circuits and the Brownian motion in stretchable crystal violet doped and biocompatible composite Hydrogels, https://doi.org/10.1016/j.molliq.2017.11.008.
The authors should present why they chose 'Electron beam radiation' as the hydrogel synthesis method, comparing it with other hydrogel synthesis methods in the introduction. They can use the following references.
1. Methods of synthesis of hydrogels … A review, https://doi.org/10.1016/j.jsps.2015.03.022.
2. Hydrogel: Preparation, characterization, and applications: A review, https://doi.org/10.1016/j.jare.2013.07.006.
3. Origin of the synthetic circuits and comparison effects of different dose malachite green oxalate doped hydrogel, https://doi.org/10.1016/j.matchemphys.2018.10.012.
It would be useful to provide more information on the physical and chemical properties of silver-doped hydrogels.
1. Silver-doped self-assembling di-phenylalanine hydrogels as wound dressing biomaterials, https://doi.org/10.1007/s10856-013-4986-2.
2. Dielectric properties of Rhodamine-B and metal doped hydrogels, http://dx.doi.org/10.1016/j.physb.2014.09.010.
Authors should achieve a standard in typeface in horizontal and vertical columns in their figures. Because let's write it in different fonts, it doesn't look good.
The characterization of the microstructure and morphology of CMCS/PVA composite hydrogel films is controversial in some more detail... 'Perhaps it has a particular impact on the higher content of PVA. Phrases such as ' should be avoided. Precise sentences may be required!
I recommend that authors make moderate English changes to this manuscript.
Author Response
Please consider as the attachment.

Reviewer 4 Report
This article entitled “Enhanced Development of Sweat Latent Fingerprints Based on Ag Loaded CMCS/PVA Composite Hydrogel Film by Electron Beam Radiation” is an interesting article reporting the preparation of Ag-loaded-CMCS/PVA (ACP) hydrogel film and the study of its use for sweat latent fingerprints investigation. The authors have used appropriate methods for their study, the text is well understood. The authors have chosen suitable and references and the English language through the whole text is correct.
Generally, I do not have many corrections to add. Some of them though, that I would like to see corrected by the authors are the following:
1) In paragraph “2.5. Morphology” the authors write “poor com-pressive strength (shown in Section 3.2). At the same time, the hydrogel has a powerful swelling capacity because of a large internal porosity, this point has been confirmed above Section 3.1.” : rephrase the text because the terms “Section 3.2” and “Section 3.1” are wrong.
2) In materials section “4.1. Materials” : rephrase the sentence “All experiments used are deionized water.”.
3) In materials section “4.1. Materials” : the authors write: Polyvinyl alcohol (PVA, 1750±50). Is it really PVA particles? Or is it a simple polymer ? What are the units of the number 1750±50 ?
4) Are the definitions of Wd and We in “equation 1” correct? I think the definition of We is wrong. If wrong, correct them.
5) In the results of Figure 3 (1), I am not sure that I see the suggestion that “The hydrogen bond force between one molecule and another molecule was reduced, and the characteristic peak at 3338 cm-1 was weakened significantly after the PVA was compounded with the CMCS”….I wish the authors could provide stronger proof.
So, after improving the context of the manuscript, I think that the present work can be published in Gels.
Author Response
Dear reviewer,
Hello! Thank you very much for your comments on this article. I have made corresponding modifications one by one. PVA(1750±50) is a product model, without punctuation, please see the following file.

Round 2
Reviewer 3 Report
manuscript is acceptable